# Development of BCMA-Targeted Bispecific Natural Killer Cell Engagers for Multiple Myeloma Treatment

**DOI:** 10.3390/antib13040097

**Published:** 2024-11-29

**Authors:** Minchuan Zhang, Han Ping Loh, Shiyi Goh Fang, Yuansheng Yang, Kong-Peng Lam, Shengli Xu

**Affiliations:** 1Singapore Immunology Network, Agency for Science, Technology and Research, Immunos Building, 8A Biomedical Grove, Singapore 138648, Singapore; zhang_minchuan@immunol.a-star.edu.sg; 2Bioprocessing Technology Institute, Agency for Science, Technology and Research, Centros Building, 20 Biopolis Way, Singapore 138668, Singapore; loh_han_ping@bti.a-star.edu.sg (H.P.L.); gohfang_shiyi@bti.a-star.edu.sg (S.G.F.); 3Department of Microbiology and Immunology, Yong Loo Lin School of Medicine, National University of Singapore, 5 Science Drive 2, Singapore 117545, Singapore; 4School of Biological Sciences, College of Science, Nanyang Technological University, 60 Nanyang Drive, Singapore 637551, Singapore; 5Department of Physiology, Yong Loo Lin School of Medicine, National University of Singapore, Medical Drive, MD9, Singapore 117549, Singapore

**Keywords:** NK cells, bispecific antibodies, multiple myeloma, functionality, format, NKCE

## Abstract

Background: B-cell maturation antigen (BCMA)-targeted T cell-redirecting immunotherapies, including Chimeric Antigen Receptor (CAR) T-cell therapy and T-cell engagers have demonstrated remarkable success in treating relapsed/refractory (RR) multiple myeloma (MM), a malignancy of plasma cells. However, a significant challenge is the severe side effects associated with T-cell overactivation, leading to cytokine release syndrome and neurotoxicity in MM patients undergoing such therapies. Bispecific NK cell engagers (NKCEs) may offer a promising alternative by redirecting NK cell cytotoxic activity towards tumor cells without triggering cytokine release syndrome. Methods: In this study, we designed a series of BCMA × CD16 NKCEs that simultaneously engage BCMA and CD16 on MM and NK cells, respectively. We evaluated the functionality of these NKCEs *in vitro* with respect to their molecular design. Results: Our results indicate that the format design of NKCEs influences their functionalities, underscoring the importance of format selection in optimizing NKCE-based therapies for MM. This study provides valuable insights for developing next-generation NKCEs and advancing therapeutic strategies for MM and potentially other malignancies.

## 1. Introduction

Multiple myeloma (MM) is the second most common hematological malignancy, which arises from the uncontrolled proliferation of plasma cells (PCs) in the bone marrow [1]. This condition manifests clinically with symptoms such as anemia, renal insufficiency, and bone damage or fractures [2]. While the average five-year survival rate for MM patients has improved to approximately 60% with standard treatments, including immunomodulatory agents, proteasome inhibitors, and monoclonal antibodies targeting CD38 or SLAMF-7 [3], many patients who initially responded to these treatments eventually become refractory to the treatment due to the development of drug resistance and subsequent relapse [4,5,6].

Recent advancements in immunotherapies, such as BCMA-targeted bispecific T-cell engagers and chimeric antigen receptor (CAR) T cells, have demonstrated remarkable efficacy in treating refractory/relapsed (RR) MM patients [7,8]. However, severe side effects associated with T cell-redirecting treatments, such as cytokine release syndrome (CRS) and immune effector cell-associated neurotoxicity syndrome (ICANS), present critical clinical challenges [9]. 

To overcome these issues, there are increasing interests in developing therapies that harness other immune cells, such as natural killer (NK) cells. NK cells can naturally destroy pathogen-infected and malignantly transformed cells without prior sensitization [10]. Unlike T cells, NK cells do not secrete high amounts of cytokines, which could induce CRS or ICANS, making NK cell-redirecting therapies safer alternatives. 

NK cell engagers (NKCEs) are innovative recombinant proteins that direct NK cells to target tumor cells [11], offering a promising direction in cancer therapy. Unlike therapies that rely on NK cells’ natural Fc receptors (e.g., CD16) or CAR-NK cell therapies, NKCEs provide unique advantages in functional flexibility and cost-effectiveness [11]. NKCEs can be tailored to bind to various activation receptors on NK cells, such as CD16, NKG2D, and Nkp46, to activate NK cells [11]. Over forty NKCEs are currently under development for the treatment of multiple cancer types [11]. Targeting CD16 is the most used strategy because it can activate NK cells independently of co-stimulation signals from other receptors, maximizing their therapeutic potential [11,12]. Clinical trials have demonstrated the efficacy of CD16-targeted NKCEs in treating both hematologic and solid tumors [13,14]. These NKCEs are either in small linked-single chain variable fragment (scFv) or large IgG-adapted format [11]. However, detailed studies on how NKCE format design affects their functionalities remain limited, and little effort has been devoted to format optimization during NKCE development.

In the current study, we investigate a series of BCMA × CD16 NKCEs designed in various IgG-adapted formats to evaluate their ability to induce NK cell-mediated toxicity against MM cells *in vitro*. Our findings demonstrate that the NKCE format plays a critical role in both manufacturability and functionality, highlighting that optimizing the format can significantly enhance therapeutic efficacy and potency. Thus, our study provides valuable insights into NKCE format optimization, paving the way for the development of next-generation NKCEs and advancing the treatment landscape for MM and potentially other malignancies.

## 2. Materials and Methods

### 2.1. Cell Culture and Plasmids

Human MM cell line KMS-11 was purchased from the American Type Culture Collection (ATCC, Manassas VA, USA). KMS-11 cell line was cultured in RPMI 1640 (Gibco, Grand Island NY, USA) supplemented with 100 U/mL penicillin (Gibco, NY, USA), 100 μg/mL streptomycin (Gibco, NY, USA), and 10% fetal bovine serum (FBS) (Hyclone, Logan UT, USA) as previously described [8]. BCMA-overexpressing HEK293T cells were purchased from (Rizhao Kanghui Biomedical Technology, Rizhao Shandong, China), and cultured in DMEM media with 10% FBS.

### 2.2. Vector Construction

The six NKCEs and two mAbs were expressed using either human cytomegalovirus promoter (hCMV)-containing or internal ribosome entry site (IRES)-containing targeting vectors. α-BCMA IgG, α-BCMA IgG (LA/LA), IgGLscFv and IgGHscFv were generated in hCMV-containing vectors as follows: α-BCMA IgG was constructed using a standard α-BCMA IgG light chain (LC) and heavy chain (HC) with α-BCMA sequence obtained from a previous study [8]. α-BCMA IgG (LA/LA) was constructed using a standard α-BCMA IgG LC and HC with LA/LA mutations at the Fc region (VH-BCMA-CHLA). IgGLscFv was constructed by linking an α-BCMA IgG LC to an α-CD16 scFv at C-terminus, combined with VH-BCMA-CHLA. IgGHscFv was constructed using a standard α-BCMA IgG LC and VH-BCMA-CHLA linked to a CD16 scFv at the C-terminus.

FabscFv, Fab2scFv, FabscFv2, and Fab2scFv2 were generated in IRES-containing vectors as follows: FabscFv was constructed using α-BCMA IgG LC and HC containing LA/LA mutations as in VH-BCMA-CHLA and engineered CH3 to form a knob (CHKLA), and an α-CD16 scFv linked to another HC with same LA/LA mutations and engineered CH3 to form a hole (FcHLA). Fab2scFv was constructed by combining an α-BCMA IgG LC, α-BCMA IgG CHKLA, and an α-BCMA VH-CH1 linked to α-CD16 scFv, which is then linked to an FcHLA. FabscFv2 was constructed using an α-BCMA IgG LC, α-BCMA VH-CH1 linked to α-CD16 scFv, which is further linked to a CHKLA and α-CD16 scFv linked to an FcHLA. Fab2scFv2 was constructed by linking an α-BCMA IgG LC and α-BCMA VH-CH1 to α-CD16 scFv, which is then linked to HC with LA/LA mutations (FcLA).

For hCMV promoter-containing constructs, five unique restriction enzyme sites (EcoRV, NotI, BamHI, NheI, and EcoRI) were included in a backbone vector. For IRES-containing vectors, four unique restriction enzyme sites (EcoRV, NotI, NruI, and EcoRI) were included between different components. Additionally, two restriction enzyme sites (MluI and MfeI) were created by mutating the six bases in front of the tenth ATG in the EMCV IRES upstream of VHBA1-CHKLA and CD16scFv-FcHLA, respectively, during gene synthesis. All vectors were generated using Genscript by inserting the genes into these unique restriction sites. The sequences of the dihydrofolate reductase (dHFR), neomycin-phosphotransferase (NPT), enhanced green fluorescence protein (EGFP), wild-type IRES, mIRES, FRT, FRT3, as well as the vector expressing enhanced FLP recombinase (FLPe), mCMV-FLPe, were described in our previous study [15].

### 2.3. Generation of Stably Transfected CHO Cell Pools and NKCE Production

CHO master cell line (MCL) cells were thawed and maintained in maintenance media for one week before transfection. The protein-free maintenance media consisted of 50% HyQ PF (GE Healthcare Life Sciences, Wilmington DE, USA) and 50% CD CHO (ThermoFisher, Waltham, MA, USA), supplemented with 1 g/L sodium carbonate (Sigma Aldrich, Darmstadt, Germany), 6 mM glutamine (Sigma), and 0.05% Pluronic F-68 (Thermo Fisher). CHO MCL cells were co-transfected with 5 µg of a vector expressing FLPe enzymes and 5 µg of the respective targeting vector using the Amaxa SG Cell Line 4D-Nucleofector^®^ X Kit L (Lonza, Basel, Switzerland). Each vector transfection was performed in duplicates. Five days post-transfection, cells were cultured in maintenance media supplemented with 20 µg/mL puromycin to select for stable cell pools. Passaging was performed every 3–4 days for two weeks until cell viability exceeded 95%.

Fourteen-day fed-batch cultures were conducted in 50 mL tube spins (TPP) within a humidified Kuhner shaker (Adolf Kühner AG, Basel, Switzerland) with 8% CO_2_ at 37 °C. Cells were seeded into 30 mL of maintenance media at an initial viable cell density (VCD) of 3 × 10^5^ cells/mL. On days 5, 7, 9, and 11, 3 mL of Ex-Cell Advanced CHO Feed 1 (with glucose) (Sigma) was added to the fed-batch cultures. 45% (*w*/*v*) d-glucose (Sigma) was supplemented as needed to maintain a glucose concentration greater than 2 g/L. VCD and viability were measured with the Vi-Cell XR viability analyzer (Beckman Coulter, Brea, CA, USA), and titre was measured using the IMMAGE 800 immunochemistry system (Beckman Coulter) on days 3, 5, 7, 9, 11, and 14 (Appendix A).

### 2.4. Purification with Protein A and SEC

Antibodies in the culture supernatant were purified using the MabSelect SuRe Protein A column (GE Healthcare) on a GE AKTA explore 100 (GE Healthcare). The protein A purified samples were analyzed by HPLC-SEC using a TSKgel G3000SWXL column (7.8 mm i.d. × 30 cm; Tosoh Bioscience) at a flow rate of 0.6 mL/min. The mobile phase comprises 50 mM MES, 200 mM L-arginine, 5 mM EDTA, and 0.05% sodium azide (*w*/*w*) at pH 6.5. The resultant concentrations were determined by comparing the area of the peaks observed at 280 nm UV absorbance to a calibration curve obtained using Trastuzumab standard samples and corrected for using the respective extinction coefficient. The calibration curve of the Trastuzumab standard used is provided in Appendix A.

### 2.5. NK Cell Isolation

Human peripheral blood mononuclear cells (PBMCs) were prepared from healthy donors using Ficoll-Paque PLUS (Cytiva, Wilmington DE, USA). Briefly, the whole blood was diluted with PBS, overlaid onto Ficoll, and centrifuged at 500× *g* for 20 min (acceleration: 9, deacceleration: 0). The PBMCs on the interface were then collected and washed with sterile PBS and subsequently used to isolate primary NK cells. Primary NK cells were isolated using a human NK cell isolation kit (Miltenyi Biotech, Bergisch Gladbach, Germany) according to the manufacturer’s protocol. The use of PBMCs from healthy donors was approved by Singhealth Centralized Institutional Review Board (CIRB). De-identified human blood tissue was collected in accordance with and under HSA Residual Blood Samples for Research, a project titled “Harnessing immune response for new therapies in transplantation and cancer” (Ref. No. 201306-04).

### 2.6. Antigen-Binding Assay

The binding of CD16 and BCMA by BCMA × CD16 bispecific NKCEs was assessed using flow cytometry using primary NK and human BCMA-expressing HEK293T cells, respectively. NK cells or HEK293T cells were plated at 1 × 10^5^/well in a 96-well U-bottom plate. To test CD16 binding, the NKCEs were individually diluted in FACS buffer (5% FBS/PBS) at various concentrations as indicated, together with a fixed concentration of α-CD16-FITC, and subsequently incubated with NK cells for 20 min. To test BCMA-binding, α-BCMA IgG control was labelled by APC using a LYNX rapid APC antibody conjugation kit (BioRad, Hercules, CA, USA). Each NKCE was diluted to its respective concentration and mixed with a fixed concentration of α-BCMA IgG-APC. The mixture would be incubated with BCMA-expressing HEK293T cells for 20 min. After washing, the cells were examined through flow cytometry. Data were collected on an LSR II flow cytometer (BD Biosciences, Franklin Lakes, NJ, USA) and analyzed with Flowjo™ v10.7 software (BD Biosciences, USA).

The percentage of binding competition (% of binding competition) is calculated by normalizing the fluorescence mean intensity (MFI) of the BCMA-HEK293T cells or NK cells by the highest and lowest MFIs, followed by the equation below:$$\%\;of\;binding\;competition\;=\;\left(\frac{{MFI}_{c}-{MFI}_{Min}}{{{MFI}_{Max}-MFI}_{Min}}\right)\times 100\%$$

${MFI}_{Max}$: MFI of antigen-positive cells with no NKCEs 

${MFI}_{c}$: MFI of antigen-positive cells at a certain concentration

${MFI}_{Min}$: MFI of Ag^+^ cells at an NKCE-saturated concentration

### 2.7. CD107a Upregulation in NK Cells

Freshly isolated human NK cells were incubated with KMS-11 in a 96-well plate at an E:T ratio of 5:1 with our BCMA × CD16 bispecific NKCEs at indicated concentrations for 4 h, with PE-conjugated α-CD107a (328608, BioLegend) at 0.7 μL per 1 million NK cells. After incubation, cells were stained with FVD510, fluorescence-conjugated α-CD56 and α-CD69 antibodies (BD Pharmingen). Samples were acquired on an LSRII cytometer (BD Biosciences) and analyzed with FlowJo 10.8.1 software.

### 2.8. In Vitro MM Cell Killing Assay

Luciferase-positive KMS-11 and purified NK cells were seeded in a 96-well plate in duplicates at an E:T ratio of 5:1 in the presence of various BCMA × CD16 bispecific NKCEs for 24 h. The viability of the KMS-11 cells was assessed using the Bright Glow Assay System (Promega, Madison, WI, USA) according to the manufacturer’s instructions. The percentage of MM cell killing by NK cells was calculated based on the difference in the loss of luminescence of the experimental well relative to the control well containing only purified NK cells with the target cells.
$$\%\;of\;killing=\frac{∆{Luminescence}_{\left(Experimental\;well-Control\;well\right)}}{Luminescence\;of\;Control\;well}\times 100\%$$

### 2.9. Cytokine Measurement by ELISA

KMS-11 and purified NK cells were cultured at E:T ratio of 5:1 in the presence of selected BCMA × CD16 bispecific NKCEs at the indicated concentrations. The culture supernatants were collected after 24 h. The NK cell-secreted cytokine IFN-γ was measured using the enzyme-linked immunosorbent assay (ELISA) using human cytokine detection kits purchased from Biolegend (San Diego, CA, USA) following the manufacturer’s protocol.

### 2.10. Detection of NKCE-Induced NK-MM Cell Doublets

Purified human NK cells were firstly stained with Cell Tracker Deep Red (Invitrogen, Waltham, MA, C34565) in PBS for 15 min at 37 °C. KMS-11-GFP (1 × 10^5^ cells) and NK cells (1 × 10^5^ cells) were cocultured at 37 °C for 2 h in a round bottom 96-well plate in the presence of different NKCE at 80 nM and 0.8 nM. Following incubation, the cells were washed on ice to remove excessive NKCEs, and the percentage of NK-MM cell doublets was assessed using flow cytometry.

### 2.11. Statistical Analysis

Statistical analysis was performed with GraphPad Prism 9 (GraphPad Software, Boston, MA, USA). A one-way ANOVA was used to compare the means between the groups. A *p*-value < 0.05 was considered significant. * *p* < 0.05, ** *p* < 0.01, *** *p* < 0.001 and **** *p* < 0.0001.

## 3. Results

### 3.1. Design of BCMA × CD16 NKCEs in Various Formats

Six different IgG-adapted bispecific NKCEs targeting human BCMA and CD16 were developed, along with two α-BCMA IgG monoclonal antibodies (mAbs) as controls, namely a wild-type α-BCMA IgG and a mutant α-BCMA LA/LA IgG, (Figure 1). The DNA sequence for α-BCMA variable (V) region of the immunoglobulin (Ig) heavy (H) and light (L) chains was obtained from our mAb specific for human BCMA, which was generated through DNA immunization as described in a previous study [8], and the DNA sequence for α-CD16 scFv was obtained from an α-CD16 mAb reported previously [16]. For all NKCEs, the BCMA-targeted arms were constructed in the Fab format, and the CD16-targeted arms were generated in the scFv format, utilizing a flexible (G_4_S)_3_ linker between the VH and VL regions. All six NKCEs contain the L234A/L235A (LA/LA) mutations in the Fc region to reduce the original Fc binding to CD16 [17], and the mutant α-BCMA LA/LA IgG carries the same mutation, whereas the wild-type α-BCMA IgG remains unaltered in the Fc region.

Among the six NKCEs, four were in IgG-like formats containing both Fab and scFv fragments, including three asymmetrical molecules (FabscFv, Fab2scFv, and FabscFv2) and one symmetrical molecule (Fab2scFv2). The remaining two NKCEs were in appended IgG formats, with scFvs linked to the C-terminus of either the IgG light chain (LC) (IgGLscFv) or the heavy chain (HC) (IgGHscFv). To enable heterodimeric Fc pairing in asymmetrical molecules, the two CH3 domains in these NKCEs were engineered to form a knob (through mutations of S354C:T366W) and a hole (through mutations of Y349C:T366S:L368A:Y407V), respectively, based on a previous study [18] (Figure 1).

### 3.2. Production of NKCEs

The six NKCEs and two α-BCMA mAbs were produced using recombinase-mediated cassette exchange (RMCE) to integrate a targeting vector expressing the specific NKCE into CHO master cell line (MCL) cells for generating stably transfected pools (Figure 2). Growth and productivity data were collected from fed-batch cultures of two independent stable pools for each NKCE (Figure 3A). These results were averaged, and the standard error of the mean (SEM) was calculated for comparison. The titers of the antibodies produced from each cell line exceeded 300 mg/L, with IgGLscFv having the highest yield at 512 ± 18 mg/L. Cell growth analysis showed that all stably transfected NKCE-producing pools had similar integrated viable cell density (IVCD) (Figure 3A), resulting in a similar trend in specific productivities (qP) for the different NKCEs (Figure 3A).

### 3.3. Comparison of Purification Efficiencies of NKCEs in Various Formats

The six NKCEs and two mAbs produced in fed-batch cultures were purified using protein A affinity chromatography. The purified products were then subjected to size exclusion chromatography (SEC) to assess aggregate levels (Table 1). The SEC profile of the eight NKCEs indicated no significant aggregation. However, we observed a low molecular weight (LMW) peak for IgGLscFv at about 23.7%, potentially indicating a drop-off of the LC-scFv (Figure 3B). Similar observations were reported previously by another group when they attached a heavy-chain-only antibody (nanobody, VHH) to the C terminal of the LC and observed an LMW peak in the SEC profile [19]. It was hypothesized that linking the VHH to the light chain weakened the association of HC and LC, resulting in the LC dropping off. Nonetheless, our results that IgGHscFv is more stable than IgGLscFv, suggesting the placement of scFv within the appended IgG format can significantly impact the stability of the product.

### 3.4. Antigen Binding Capacities of NKCEs in Different Formats

The binding of the NKCEs to both BCMA and CD16 is essential to direct NK cells against BCMA-expressing MM cells. As the valency of our NKCEs against each antigen, BCMA or CD16, are different, direct binding assays using an α-Fc antibody would result in an inaccurate measurement. To solve this issue, we utilized a competitive binding assay to determine the respective binding of each NKCE towards BCMA or CD16. 

First, to test the BCMA-binding of NKCEs, α-BCMA IgG (Figure 1) was conjugated with allophycocyanin (APC) using an antibody conjugation kit. Subsequently, human BCMA-overexpressing HEK293T cells were incubated with the α-BCMA IgG-APC at a fixed concentration in the presence of various NKCEs in varying concentrations. In this experimental setting, the APC fluorescence intensity is reversely correlated to the BCMA-binding of the NKCEs on the HEK293T cells. The percentage of binding competition was calculated according to the equation in Materials and Methods. A lower concentration of an NKCE required for the same percentage of binding competition indicates a stronger BCMA-binding affinity of the NKCE (Figure 4A). We observed that FabscFv and FabscFv2 exhibited the weakest binding to BCMA on HEK293T cells, while the other four NKCEs and the two mAbs showed largely comparable BCMA-binding capabilities. The FabscFv and FabscFv2 NKCEs possess only one BCMA-binding moiety, whereas the other four NKCEs and the two mAbs have two BCMA-binding moieties (Figure 1). These results suggest that the BCMA-binding capabilities of NKCEs are influenced by the valency of the BCMA-binding moiety. 

Next, we examined the CD16-binding using a similar approach. We selected a commercial α-CD16-FITC antibody (Clone B73.1) that could compete with the NKCEs for CD16-binding. Using this α-CD16 mAb at a fixed concentration, we performed CD16 competitive binding assays with primary NK cells, varying the concentrations of the different NKCEs. We first observed that α-CD16 scFv of NKCEs exhibited stronger binding to CD16 than the Fc region of the α-BCMA IgG (Figure 4B). The LA/LA mutation in the Fc region significantly reduced CD16-binding, as α-BCMA IgG LA/LA failed to compete with the α-CD16-FITC antibody for CD16-binding (Figure 4B). Although both FabscFv and Fab2scFv are monovalent for CD16-binding, the former exhibited much stronger CD16-binding than the latter, likely due to hindrance from the α-BCMA Fab at the N-terminus of the scFv. 

Amongst the four NKCEs that are bivalent for CD16-binding—IgGLscFv, IgGHscFv, Fab2scFv2, and FabscFv2—IgGLscFv displayed the weakest CD16-binding, even lower than α-BCMA IgG (Figure 4B,C). This indicates that appending scFv at the C-terminus of the light chain impedes CD16-binding, correlating with its weaker molecular stability (Figure 3B). On the other hand, FabscFv2 exhibited much stronger CD16-binding than Fab2scFv2 (Figure 4C), similar to the pattern observed with FabscFv and Fab2scFv (Figure 4B). In fact, FabscFv2 demonstrates the strongest CD16-binding among the six NKCEs.

### 3.5. Functional Characterization of NKCEs in Various Formats

We next assessed the functionalities of these NKCEs *in vitro*. First, we evaluated their capability to induce NK cell cytotoxicity against BCMA-expressing MM cells. Freshly purified NK cells from PBMCs were co-cultured with a luciferase-positive MM cell line, KMS-11, at a 5:1 effector (E)/target (T) ratio in the presence of various NKCEs at indicated concentrations for 24 h. The luminescence intensity in the culture was used as a surrogate for KMS-11 cell viability, allowing us to determine the percentage of the specific lysis of MM cells. 

It was found that IgGHscFv NKCE possesses the strongest killing potency across all three concentrations tested. At a concentration of 0.008 nM, it already could achieve 60–70% killing (Figure 5A), whereas the other four NKCEs—Fab2scFv, FabscFv, Fab2scFv2, and FabscFv2—only reached 10–20% killing, similarly to the α-BCMA IgG. At the same time, IgGLscFv NKCE along with α-BCMA IgG (LA/LA) failed to induce NK cell cytotoxicity at this concentration. When the concentration was increased to 0.8 nM, the killing potencies of the aforementioned four NKCEs rose to 40–80%, still slightly lower than the 90% killing potency of IgGHscFv NKCE (Figure 5B). At a higher concentration of 80 nM, the killing potencies of FabscFv and FabscFv2 became comparable to that of IgGHscFv. At this concentration, the remaining three NKCEs—Fab2scFv, IgGLscFv, and FabscFv—achieved 40–60% killing (Figure 5C). These results suggest that IgGHscFv is a promising BCMA-targeted NKCE candidate for future development due to its high potency of inducing NK cell-mediated MM cell lysis at very low concentrations, outperforming the other NKCEs and α-BCMA IgG.

We further evaluated the *in vitro* functionality of the NKCEs by assessing their ability to induce NK cell degranulation against KMS-11 cells. NK cells were incubated with KMS-11 cells at a 5:1 E/T ratio for 4 h in the presence of various NKCEs at different concentrations. The degranulation of NK cells induced by various NKCEs was assessed by measuring CD107a upregulation on NK cells using flow cytometry (Figure 6A). The upregulation of CD107a has been demonstrated to be strongly correlated with NK cell activation [20]. Consistent with the cytotoxicity results, IgGHscFv was among the best three NKCEs—alongside FabscFv and FabscFv2— inducing CD107a upregulation. Moreover, IgGHscFv was the most effective NKCE at inducing CD107a upregulation at concentrations below 0.8 nM (Figure 6B). While FabscFv and FabscFv2 were also strong inducers of CD107a upregulation at higher concentrations (0.8 nM or greater), their effectiveness diminished quickly at lower concentrations (Figure 6B). The overall trend of NKCE-mediated NK cell degranulation was consistent with their ability to induce specific lysis of MM cells (Figure 6C). 

We also examined IFN-γ production by NK cells induced with various NKCEs. IgGHscFv was found to be the most potent NKCE in inducing IFN-γ production, followed by FabscFv, FabscFv2, Fabs2scFv, and FabsscFv2. The overall trend in the NKCEs’ ability to induce IFN-γ production closely mirrored their ability to induce NK cell degranulation (Figure 6D). Together, our results suggest that IgGHscFv is the most effective NKCE among the six we designed to target BCMA-expressing MM cells, as it exhibited the strongest killing potency against MM cells, as well as the strongest induction of NK cell degranulation and IFN-γ production.

### 3.6. Immunological Synapse Formation Induced by Various NKCEs 

The classical immunological synapse refers to the interface between an antigen-presenting or target cell and a T cell [21]. NK cells are also known to form immunological synapses with cytolytic effects against the target cells, such as tumor cells, facilitating the release of lytic granules, including perforin, granzymes and lysosomal hydrolases into the tumor cells to induce cytolysis [22]. Similar to T cell engagers, which can meditate the formation of immunological synapses [23], NKCEs can also elicit this effect to kill the target cells. To further explore the mechanistic insights into how different NKCE designs impact their cytotoxicity, we attempted to assess the formation of immunological synapses induced by various NKCEs.

In the initial step of immunological synapse formation, NK cells approach the target cells and form a conjugate through direct interaction of the surface molecules expressed on both NK and target cells, followed by specific signaling-mediated accumulation and polarization of actin, leading to the formation of large multi-molecular complexes [22]. Therefore, we next used the formation of NK-MM cell conjugate as a surrogate to assess the ability of various NKCEs to induce the formation of immunological synapses using a flow cytometric approach as described previously [24].

NK cells were firstly labelled by Cell Tracker Deep Red and then co-cultured with GFP^+^ KMS-11 at a 1:1 E/T ratio for two hours in the presence of various NKCEs at 80 or 0.8 nM. The formation of NK-MM cell conjugate was assessed by the percentage of NK and KMS-11 cell doublets, which are positive for the GFP and Deep Red as measured using flow cytometric analysis. We observed that FabscFv2, FabscFv, IgGHscFv, and FabscFv2 NKCEs induced approximately 15% doublets (Figure 7). In contrast, the percentages of NK-MM cell doublets induced by IgGLscFv and Fab2scFv NKCEs were only around 6–7%, significantly lower than the former four NKCES. When correlated with the killing of MM cells (Figure 5A), the ability of these NKCES to induce the formation of NK-MM cell doublets was largely consistent with their cytotoxic effectiveness (Figure 5A and Figure 7B). These results suggest the strong capability of an NKCE to induce immunological synapse formation is critical for its cytotoxic efficacy.

## 4. Discussion

In this study, we designed and generated six BCMA × CD16 NKCEs in various formats and systemically evaluated the impact of format design on their developability and functionality. Specifically, we assessed their yields, structure stability, antigen-binding capacity, and MM cell-specific cytotoxicity. 

Our study demonstrates that structure stability is critical for the functionality of NKCEs. Although the yields of the NKCEs are comparable, IgGLscFv displays poor structural stability (Figure 3B). Subsequent functional assessments have revealed that this instability compromised its ability to bind antigens, specifically lyse MM cells, induce NK degranulation, and mediate immunological synapse formation. Conversely, IgGHscFv, which has the same antigen valencies as IgGLscFv, demonstrates good structure stability and has excelled in all functional assessments. The location of the scFv appendage may also introduce steric hindrance, affecting its ability to bind to CD16. However, a T-cell engager with a similar IgGLscFv format has demonstrated satisfactory efficacy against tumor cells [25], suggesting that the engagement of CD3 on T cells and CD16 on NK cells using bispecific antibodies in the same format might differ. 

Our findings also reveal that the functionality of NKCEs is not only determined by their antigen-binding abilities. For example, while IgGHscFv and Fab2scFv2 exhibit similar binding abilities to BCMA and CD16, IgGHscFv significantly outperforms Fab2scFv2′s potency to kill MM cells. This suggests that additional factors are influencing the efficacy. Another example involves FabscFv and FabscFv2; although FabscFv demonstrates weaker binding to CD16 than FabscFv2, both NKCEs exhibit similar efficacy in inducing KMS-11-specific lysis. These findings underscore the complex interplay of multiple functional attributes, not just antigen binding, in determining the overall therapeutic effectiveness of NKCEs. Similar observations have been reported in certain T-cell engagers, where the antigen-binding strength does not necessarily correlate with their efficacy [26,27]. 

Furthermore, our results underscore the importance of NKCEs’ ability to facilitate immunological synapse formation, which is crucial for their functional effectiveness, similar to the role observed in CD8^+^ T cells [28]. Our results demonstrate a clear correlation between the ability of NKCEs to induce the formation of immunological synapses between NK and target cells and their effectiveness in mediating KMS-11 lysis and NK cell degranulation. Notably, IgGHscFv NKCE stands out for its superior ability to initiate immunological synapse formation, resulting in the most efficient killing of MM cells (Figure 7). On the other hand, Fab2scFv and IgGLscFv show the weakest capability to induce immunological synapse formation, correlating with their lower efficacy in tumor cell eradication. These findings indicate that the ability of NKCEs to induce immunological synapse formation is paramount for their anti-tumor functionalities.

Interestingly, our results show that α-BCMA IgG induces significant specific killing of KMS-11 by NK cells, despite its lower binding to CD16 and its reduced capacity to activate NK cells (Figure 4B and Figure 6C). One possible explanation for this phenomenon is that the weaker binding of α-BCMA IgG to CD16 through its Fc region facilitates rapid disengagement of NK cells after delivering a lethal hit to target cells, potentially allowing a single NK cell to lyse multiple target cells [29]. Similarly, a previous study also showed that CAR-T cells with CARs of moderate binding affinity to tumor antigens exhibited superior clinical response compared to those with higher affinity [30]. However, further experiments are necessary to validate this hypothesis.

## Figures and Tables

**Figure 1 antibodies-13-00097-f001:**
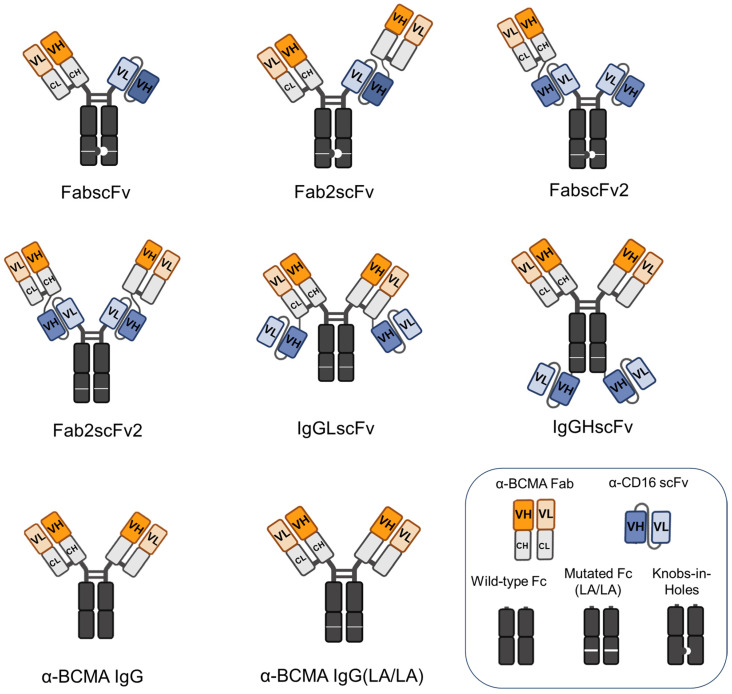
Schematic diagram of the formats of BCMA × CD16 NKCEs. The six NKCEs targeting BCMA and CD16 are designed differently to test the effects of valency and orientation on therapeutical potential. **Top**: The three NKCEs are asymmetrically designed with a valency of “1+1”, “2+1” and “1+2” against BCMA and CD16, respectively. **Middle**: These three NKCEs are symmetrically designed with α-CD16 scFv at different positions of light chains and heavy chains. **Bottom**: Anti-BCMA IgG and α-BCMA IgG (LA/LA) are used as controls. α-BCMA IgG (LA/LA) has a mutated Fc domain that minimizes its binding to CD16 compared to anti-BCMA IgG.

**Figure 2 antibodies-13-00097-f002:**
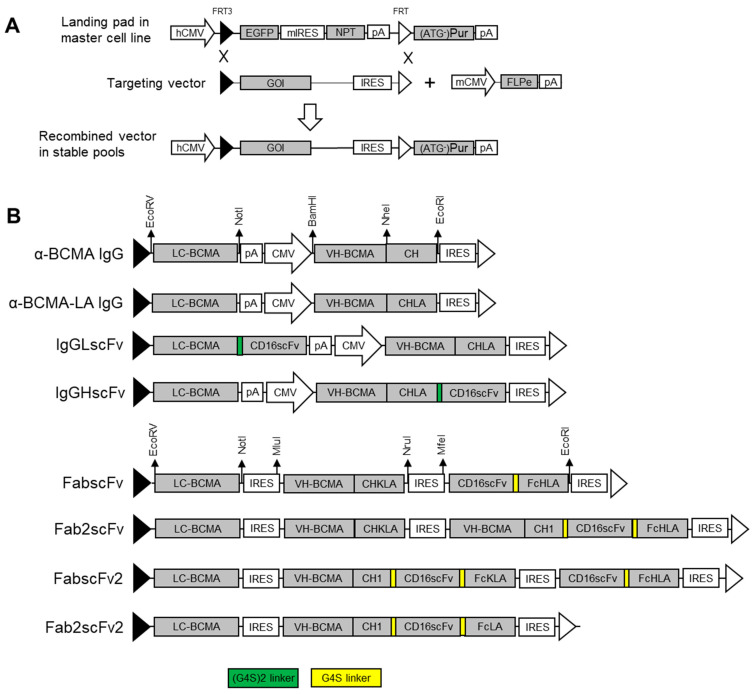
Schematic diagram of the RMCE process and targeting vector design (**A**) Overview of the RMCE targeted integration platform for expressing antibodies in CHO cells. CHO-K1 cells were stably tagged with a Flp/FRT RMCE cassette (landing pad) via random integration. The targeting vector was promoter-less, containing FRT3/FRT sites, the gene of interest (GOI), and an IRES element to activate puromycin gene expression upon successful RMCE. (**B**) Schematic representation of targeting vectors carrying antibody genes linked by CMV promoter or multiple wild-type EMCV IRES elements.

**Figure 3 antibodies-13-00097-f003:**
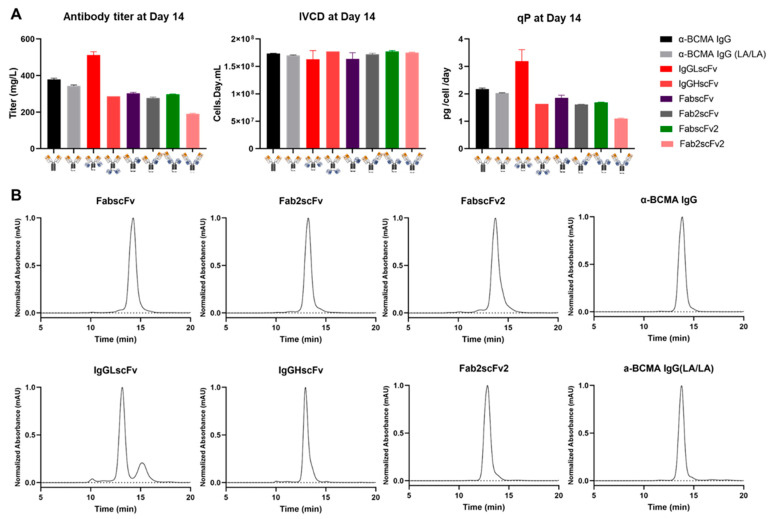
Production and purification of NKCEs. (**A**) Titer, IVCD, and qP of NKCEs in culture. Error bars show the SEM of duplicate fed-batch cultures. IVCD: Integral viable cell concentration; qP: specific productivity, calculated using day 14 titer over day 14 IVCD. (**B**) SEC profile of different antibodies after protein A purification.

**Figure 4 antibodies-13-00097-f004:**
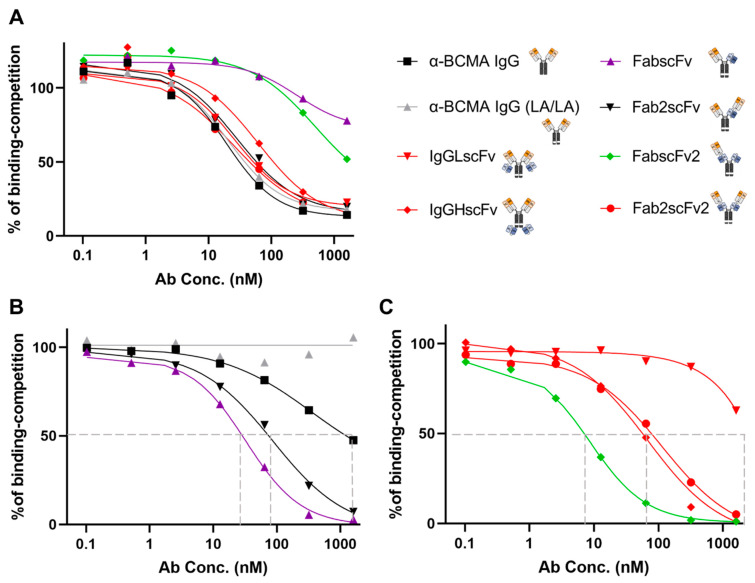
Flow cytometric analysis of BCMA and CD16 binding capacities of BCMA × CD16 NKCEs. (**A**) BCMA binding capacities of the NKCEs were tested using a competitive binding assay through flow cytometry. BCMA^+^ HEK293T cells were incubated with serially diluted NKCEs and a fixed concentration of anti-BCMA IgG conjugated with APC. FabscFv and FabscFv2 show weaker binding to BCMA, while the rest of NKCEs and mAbs show comparable binding capacities to BCMA. (**B**,**C**) CD16 binding capacities of NKCEs were tested using a competitive binding assay through flow cytometry. Primary NK cells were incubated with the serially diluted NKCEs and a fixed concentration of a commercial anti-CD16-FITC antibody (BD Pharmingen). The IC50 value of the NKCEs and mAbs are as follows: FabscFv (IC50 = 32.52 nM), Fab2scFv (IC50 = 86.73 nM), anti-BCMA IgG (IC50 > 1000 nM), anti-BCMA IgG (LA/LA) (IC50 > 1000 nM), IgGLscFv (IC50 > 1000 nM), IgGHscFv (IC50 = 68.38 nM), and Fab2scFv2 (IC50 = 114.3 nM).

**Figure 5 antibodies-13-00097-f005:**
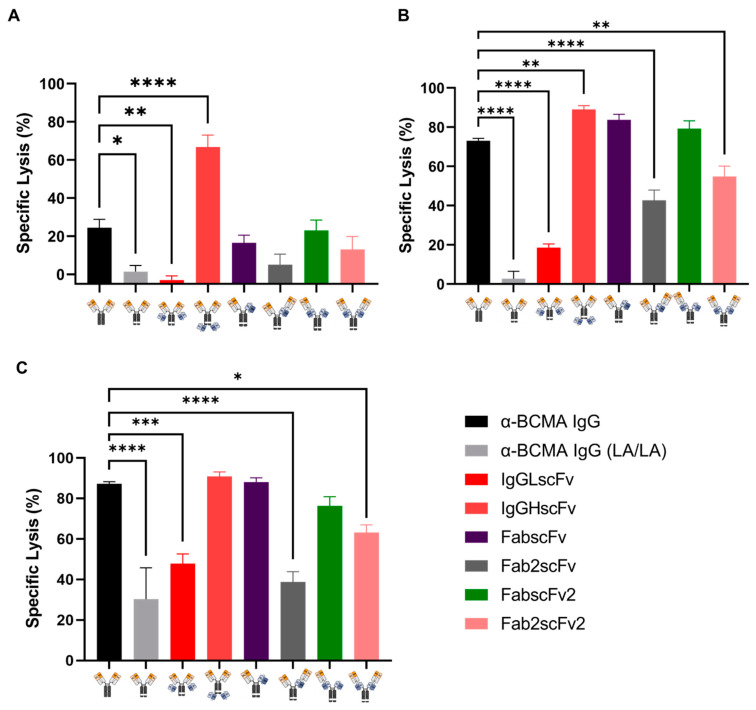
Tumor cell killing using NK cells induced by BCMA × CD16 NKCEs. The NKCE-mediated killing of KMS-11 using NK cells was examined at different NKCE concentrations. Purified primary NK cells were co-incubated with BCMA^+^ KMS-11-luciferase cells (E:T = 5:1) in the presence of individual NKCEs at the indicated concentration for 24 h. The luciferase activity was used as the surrogate for cell viability and determined by measuring the luminescence in duplicates after adding the substrate luciferin. (**A**) Quantitative comparison of specific lysis of KMS-11 by individual NKCE at 0.008 nM, (**B**) 0.8 nM, and (**C**) 80 nM (n = 5). * *p* < 0.05, ** *p* < 0.01, *** *p* < 0.001 and **** *p* < 0.0001.

**Figure 6 antibodies-13-00097-f006:**
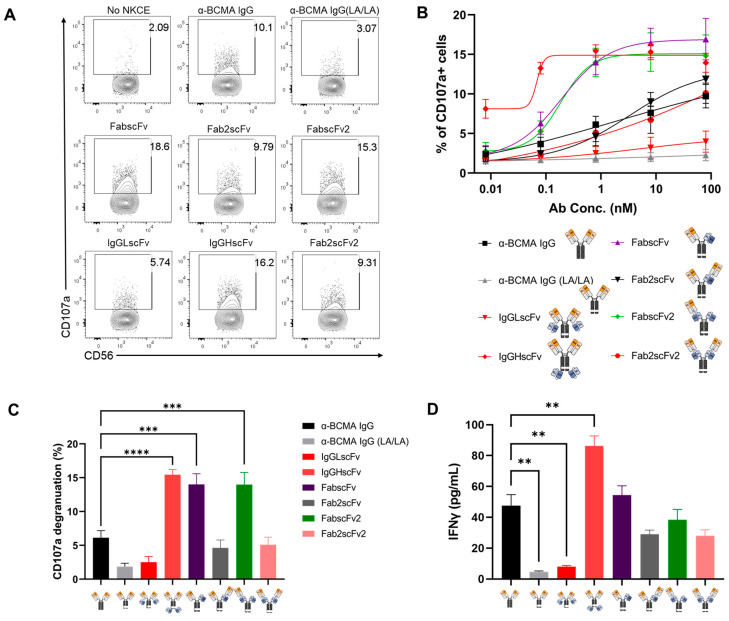
NK cell degranulation upon stimulation by various NKCEs and cytokine production in the presence of tumor cells. (**A**) Representative flow cytometric plot of degranulation marker CD107a on NK cells co-cultured with KMS-11 for 4 h in the presence of various NKCEs at 80 nM. (**B**) Summary of CD107^+^ NK cells after 4-h incubation with KMS-11 in the presence of different NKCEs at varying concentrations (n = 5). (**C**) Quantitative comparison of NK degranulation induced by different NKCE at 0.8 nM. (**D**) IFNγ cytokine production by NK cells. The supernatant was harvested after 24-h co-incubation of NK cells and KMS-11 with NKCEs at 80 nM. IFNγ was measured by ELISA. ** *p* < 0.01, *** *p* < 0.001 and **** *p* < 0.0001.

**Figure 7 antibodies-13-00097-f007:**
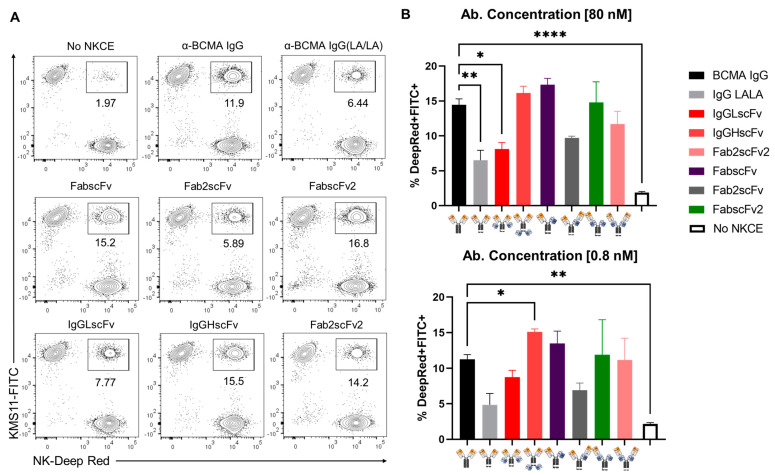
Formation of NK-KMS-11 cell aggregates mediated by individual NKCEs. GFP^+^ KMS-11 and purified human NK cells were co-cultured (E:T = 1:1) in the presence of different NKCEs (80 nM or 0.8 nM) at 37 °C for 2 h. NK cells were labelled by Cell Tracker Deep Red. After incubation, (**A**) FACS measurement was used to determine the percentage of NK and KMS-11 cells forming aggregates induced by NKCEs (FITC^+^ Deep Red^+^ population). (**B**) Summary of the double positive populations from multiple independent experiments (mean ± SEM). * *p* < 0.05, ** *p* < 0.01 and **** *p* < 0.0001.

**Table 1 antibodies-13-00097-t001:** Product quality results for varying vector configurations.

Format	Post Protein A Purification		
	SEC Analysis		
	HMW (%)	POI (%)	LMW (%)
α-BCMA IgG	0.60	97.83	1.56
α-BCMA-LA IgG	0.62	97.34	2.04
IgGLscFv	4.81	71.46	23.73
IgGHscFv	3.14	95.85	1.01
FabscFv	3.27	95.30	1.42
Fab2scFv	2.70	94.33	2.97
FabscFv2	3.51	92.95	3.54
Fab2scFv2	0.28	96.71	3.01

SEC: size-exclusion chromatography; HMW: high molecular weight; POI: product of interest; LMW: low molecular weight.

## Data Availability

Original data for this study could be requested by email to the corresponding authors.

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
