# Peer review of "Development of BCMA-Targeted Bispecific Natural Killer Cell Engagers for Multiple Myeloma Treatment"

_2073-4468, 2024, doi:10.3390/antib13040097_

Round 1

Reviewer 1 Report

Comments and Suggestions for Authors

-The authors use a cellular aggregation assay to measure immunosynapse formation. It is important to reference the paper that originally described the assay development. This is also true for the CD107a assay. 

-The purpose of a manuscript discussion is for authors to discuss their findings in relation to the finding of other authors in the literature. A major concern is that, in the discussion, the authors mostly discuss their own data and reference the work of few others. 

-Did the authors create and study any additional molecules as negative controls? (perhaps creating a CD16/CD19 IgGHscFv. Negative subclass controls are quite useful to further establish specificity.  If not, why?

-It would be helpful to define all abbreviations. Especially those used in Table 1. 

-Please clearly state the original contributions of this work to the field of study. 

-Why did the authors not include any in vivo work in mice. It would be very important to know the stability/clearance/pk of these molecules in vivo? 

Reviewer 2 Report

Comments and Suggestions for Authors

This study presents NK cell engagers as a promising therapeutic approach for multiple myeloma, focusing on a series of BCMA×CD16 NKCEs that engage BCMA on MM cells and CD16 on NK cells simultaneously. the findings offer evidence of the potential advantages of NKCEs over traditional T-cell-based immunotherapies. While the addition of in vivo studies would strengthen the clinical relevance and impact, the results here provide a basis for future research.

Comments:

The authors should include the cytotoxicity curve of puromycin treatment used for selecting stably transfected CHO cell pools

Some data points lack detailed statistical analysis (e.g., Figures 6D and 7B). Adding statistical details would improve the rigor and interpretation of the results.

More details are needed regarding the antibody purification via HPLC, limits of detection, limits of quantification, and calibration curve, R2 , etc. These details will enhance the reproducibility of the purification process described by the authors

Round 2

Reviewer 1 Report

Comments and Suggestions for Authors

My concerns have been addressed. However, I still feel that with minimal in vivo studies, the paper would be far more interesting and important. However, the in vitro comparison is still interesting.